REGISTERED REPORT PROTOCOL

# Does darkness increase the risk of certain types of crime? A registered report protocol

Jim Uttley[1]*, Rosie Canwell[2], Jamie Smith[2], Sarah Falconer[2], Yichong Mao[1], Steve A. Fotios[1]

1 School of Architecture, University of Sheffield, Sheffield, United Kingdom, 2 South Yorkshire Police, Sheffield, United Kingdom

* j.uttley@sheffield.ac.uk

**Data Availability Statement:** The fictional data that has been used to demonstrate our proposed

## Abstract

Evidence about the relationship between lighting and crime is mixed. Although a review of evidence found that improved road / street lighting was associated with reductions in crime, these reductions occurred in daylight as well as after dark, suggesting any effect was not due only to changes in visual conditions. One limitation of previous studies is that crime data are reported in aggregate and thus previous analyses were required to make simplifications concerning types of crimes or locations. We will overcome that by working with a UK police force to access records of individual crimes. We will use these data to determine whether the risk of crime at a specific time of day is greater after dark than during daylight. If no difference is found, this would suggest improvements to visual conditions after dark through lighting would have no effect. If however the risk of crime occurring after dark was greater than during daylight, quantifying this effect would provide a measure to assess the potential effectiveness of lighting in reducing crime risk after dark. We will use a case and control approach to analyse ten years of crime data. We will compare counts of crimes in 'case' hours, that are in daylight and darkness at different times of the year, and 'control' hours, that are in daylight throughout the year. From these counts we will calculate odds ratios as a measure of the effect of darkness on risk of crime, using these to answer three questions: 1) Is the risk of overall crime occurring greater after dark than during daylight? 2) Does the risk of crime occurring after dark vary depending on the category of crime? 3) Does the risk of crime occurring after dark vary depending on the geographical area?

## 1. Introduction

Darkness is likely to influence perceptions of safety and fear of crime amongst pedestrians. Previous studies have found significant differences in subjective evaluations of safety and fear of crime under different conditions of ambient light–i.e. daylight versus darkness (e.g. 1–3). These differences have been found in a range of different types of study design. For example, Gover et al [1] asked staff and students on a University campus about their fear of crime and perceived risk of crime during the day and at night. Fear of crime and perceived risk of crime were higher at night than during the day. Some studies show participants photographs or

analysis is included as Supplementary files S3 and S4. It will not be possible to make openly available the real raw crime data that will be used. These data will include details of the category of crime, the time and date the crime occurred, and its location. These details could potentially allow the identification of individuals, hence why the raw data cannot be made available. We will instead provide aggregated counts of crimes for the CaseDark, CaseDay, ControlDark and ControlDay periods, by crime category, and by MSOA.

**Funding:** The authors received no specific funding for this work.

**Competing interests:** The authors have declared no competing interests exist.

virtual reality images representing daylight and after dark conditions, and ask about perceptions of safety. In one study, Loewen, Steel & Suedfeld [2] showed participants a series of photographs of scenes that were in daylight or darkness. Daylight scenes produced significantly higher ratings of safety than those scenes shown in darkness. Other studies have taken participants to real locations in both daylight and after dark and asked them to evaluate how safe they felt in that location. Boyce et al [3] took participants to 24 open parking lots and found that ratings of how safe it felt to walk at those parking lots were consistently lower during the after-dark visit than during the daylight visit. A similar method was used by Fotios, Monteiro & Uttley [4] who took participants to ten locations in an urban residential area. Ratings of safety during the after-dark visits were significantly lower than during the daylight visits. Research also shows a relationship between the level of light provided by street lighting after dark and feelings of safety. The study by Boyce et al [3] for example showed that as the median illuminance of a car park increased the difference between daylight and after dark ratings of safety reduced, suggesting participants felt safer as illuminance increased. The study by Fotios, Monteiro & Uttley [4] demonstrated a similar relationship between illuminance and feelings of safety.

In summary, a range of studies using different methods have demonstrated perceptions of safety tend to be lower (and fear of crime higher) after dark than during daylight, and when an area is less well lit after dark. One reason why it feels less safe after dark is that visual function is impaired at low light levels [5] meaning it becomes more difficult to detect and identify visual features of the environment that might contribute to judgements of prospect and refuge, which are known to influence how safe we feel [6].

In contrast to perceived risk of crime and perceptions of safety, the relationship between light levels and actual crime and safety is less clear. Evidence about offender perceptions and their decision-making processes does not reveal a consistent influence of darkness or daylight–this influence is mediated by other factors such as victim type and type of crime. In interviews with sexual assault offenders, Balemba and Beauregard [7] found that offenders were more likely to commit a sexual assault after dark if their victim was an adult, but a younger victim was more likely to attacked during daylight. Palmer, Holmes and Hollin [8] found that domestic burglars had no clear preference for when they committed their offence–a third preferred the morning (presumably in daylight although this is not stated) and a third preferred after-dark. In contrast, Tabirizi and Madanipour [9] found that domestic burglars in Tehran did prefer to commit burglary after dark as it reduced the chance of being seen by neighbours and passers-by, and made it easier to see whether occupants were at home or not. Street robbers might be expected to prefer to commit robbery after dark, as there are fewer people around to intervene and they will be less identifiable to their victim, but interviews with convicted street robbers suggested there was no general preference for committing robbery after dark–offenders tended to be active at all times of the day [10]. Findings from interviews with offenders do not therefore suggest a clear preference for committing crime after dark, and if there is any preference this is likely to be mediated by other factors such as the type of crime and victim characteristics.

Evidence from data about committed crime also fails to reveal a clear pattern in relation to light conditions. Steinbach et al [11] examined how Local Authority street lighting strategies for energy saving such as dimming and partial switch-off influenced crime counts. They found no clear relationship between any of the lighting strategies examined and the numbers of crimes recorded. The study has limitations however that raise a question over this conclusion (e.g. see [12]). For example, only 62 out of 174 Local Authorities in England and Wales provided data for the study: A self-selection bias may have occurred, with Local Authorities more likely to provide their data if they anticipated no detrimental effect of lighting switch-off or

dimming on crime rates. The crime data used in the study did not include information about the time of occurrence meaning it was not possible to distinguish between crimes that occurred after dark or during daylight. In an attempt to address this, the authors of that work included only crime types that were believed more likely to occur after dark (e.g. burglary, theft from vehicle, robbery, sexual assault). Further analysis of the relationship between switching off street lights and crime rates found that theft from vehicle offences reduced when street lights were switched off after midnight, an opposite effect to the assumed benefit lighting has on the occurrence of crime [13]. Spatial displacement was found, with an increase in theft from vehicles in adjacent areas where street lighting remained unchanged, further evidence that the presence of lighting could increase rather than decrease certain crimes. Davies and Farrington [12] examined the effect of switching off street lighting at night (between 23:30 and 05:30) on crime. They compared one district that carried out this switch-off with another district that did not. Results were mixed–burglary and vehicle crime reduced in the switch-off district but not as much as in the control district; however, violent crime reduced significantly more in the switch-off district compared with the control district.

A systematic review of the effect of lighting interventions on crime rates found that improved street lighting (for example newly-installed or brighter lighting) was associated with a reduction in crime [14]. However, reductions in crime were found in daylight as well as after dark, suggesting any effect of the lighting was not due to improvements in visibility in an area but instead to an increase in 'community pride' - an area receiving local government attention in the form of street lighting improvements prompts increased community cohesiveness, informal social control and more people outdoors, with more eyes on the street being the cause of a reduction in crime. The review by Welsh and Farrington has been criticised on statistical grounds [15, 16]. One example of such criticism is that areas may receive investment to improve lighting because they have high levels of crime. Regression to the mean implies that crime levels would have reduced anyway in these areas, regardless of any lighting intervention. Welsh and Farrington recently updated their review of the effect of street lighting on crime rates [17]. This review included the original thirteen studies included in their earlier review along with eight new studies published since the earlier review that met the inclusion criteria. The new review concluded that there is still a desirable effect of street lighting on crime based on the evidence, with crimes decreasing by 14% in treatment areas that receive improved street lighting compared with control areas that do not. However, this includes crimes committed during daylight as well as after dark. When only studies that included after-dark crime in their analyses were included in the meta-analysis, a non-significant 3% decrease in crimes in treatment areas was found.

One of the key studies included in the second Welsh et al review [17] was a randomised controlled trial of the effect of street lighting on crime [18], the first of this type of study on the topic. Mobile lighting towers were deployed in 40 randomly-selected housing developments and crime counts were compared with 40 control housing developments where the lighting towers were not deployed. Both groups of housing developments had high baseline levels of crime which helped overcome the issue of regression to the mean highlighted by Marchant [15]. Results suggested the lighting intervention led to a 35% reduction in serious crime after dark in the 6 months after the lighting deployment. It is not known whether this reduction in crime continued after 6 months though. A further aspect of the study to note is that the lighting intervention used in this study was not typical street lighting but mobile towers powered by a diesel generator. Each tower provided 600,000 lumens–between 17 and 120 times greater than the lumens provided by existing outdoor lighting in the housing developments. They therefore provided better visibility and were more obvious interventions than typical street lighting.

The relationship between street lighting and crime rates continues to remain unclear–previous studies show mixed or weak results, and have a range of limitations. To help clarify the relationship between lighting and crime rates it is first useful to understand whether extreme differences in light levels - daylight versus darkness - produce differences in crime. If such an extreme difference in light levels shows no influence on crime rates, it seems unlikely that smaller changes in light levels brought about through lighting strategies or interventions can have any significant impact on crime rates. A comparison of ambient light conditions (daylight and darkness) can be done when there is no lighting intervention involved. Therefore, if any effect is found, it could be attributed to changes in light levels rather than changes in community pride caused by investment in an area. Distinguishing between the crime reducing effects of visibility and community pride is useful because there may be better ways than improving street lighting for enhancing community pride, perhaps methods which do not consume energy in operation or impact upon the nocturnal environment. The size of any effect found can also provide a useful measure of the effectiveness of any outdoor lighting strategy to reduce crime after dark. Effective lighting for crime reduction would reduce the risk of crime occurring after dark compared with during daylight in an area.

To measure the effect of darkness and lighting on crime it is necessary to account for other contributory factors that may act as confounds. It is insufficient to simply compare crime counts during periods of daylight with periods of darkness as a range of other confounding factors will also influence crime occurrence in addition to any effect of the ambient light condition. These include environmental factors such as shop density and house prices [19], socio-economic factors such as unemployment rates and income levels [20], and temporal factors such as time of day (e.g. [21]) and seasonal changes in weather conditions (e.g. [22, 23]) which may affect the routine activities of people and motivation of offenders. For example, Herrmann [24] describes daily temporal patterns for murders, shootings, assaults and robberies in the Bronx, New York. There are large differences in the numbers of crimes depending on the time of day, with late evening generally showing peak crime counts. However, the peak hour for robberies is 15:00. Schools tend to finish around this hour, and further analysis of crime records showed that this hourly peak in robbery was largely caused by robberies involving school-aged offenders near schools and subway stations. These results show how the opportunity and motivation for crime vary depending on the time of day, as would be predicted by routine activity theory [25]. This highlights how the time of day influences the likelihood of crime occurring, including the type of crime that is committed.

One approach to overcoming confounding factors such as time of day and weather conditions is to use daylight saving time clock changes to create a natural experiment where time of day and other seasonal influences can be controlled whilst varying the ambient light. At a Spring clock change, clocks are moved forward one hour, leading to a sudden, additional hour of daylight in the evening (whilst an hour of daylight in the morning is lost). At an Autumn clock change this is reversed, with an evening hour of daylight being lost as clocks are shifted back one hour. Comparing crime rates in the weeks immediately before and after a clock change, particularly around the hour of sunrise or sunset, can say something meaningful about the effect of ambient light conditions. Doleac and Sanders [26] carried out such an analysis of crime records in the United States. The shift from darkness to daylight in the sunset hour following the Spring clock change was associated with a 27% decrease in robberies and a 38% decrease in rapes. This study primarily analysed crime data from less populous areas however, and the timing of crimes may vary between rural and urban areas.

The one-hour shift in evening daylight caused by the daylight saving time clock changes was also exploited by Fotios, Robbins and Farrall [27] to investigate the influence of ambient light on crime counts. They used a case / control method to isolate the effect of darkness on crime

counts. A 'case' window of time was selected that was in darkness before a clock change but was in daylight after the clock change (for a Spring clock change; this was reversed for an Autumn clock change). Counts of crimes that occurred in these two windows were compared in the week before and after the clock change. The ratio of counts between the after-dark and daylight case windows were then compared to a ratio of counts over a similar period but in control windows - periods of time that had the same duration as the case window but that occurred either two hours earlier or later so they had the same light condition (daylight or darkness) both before and after the clock change. The comparison of these two ratios was calculated as an odds ratio, where an odds ratio significantly greater than one would indicate crime was more likely to occur after dark than during daylight. Fotios, Robbins and Farrall [27] applied this method to crime data from three cities in the United States for the period 2010–2019. Results suggested a statistically significant effect of darkness on robbery but not on other types of crime. Fotios, Robbins and Farrall [28] extended this work by examining the influence of ambient light on crime in eleven cities in the United States. This further confirmed a consistent effect of light on robbery, with this type of crime increasing when it was dark, but no other significant effects were found for other crime types. An effect of darkness on robbery has also been shown in other work that compared counts of robberies in 6-hour periods of the day that varied in terms of the proportional amount of darkness they contained [29].

The case / control method used by Fotios, Robbins and Farrall [27, 28] helps to isolate the effect of darkness by controlling for other confounding factors that are likely to influence crime rates, such as the time of day and weather conditions. However, one limitation of using the clock change to produce a sudden shift in ambient light conditions is the period of time in which crimes are included in the analysis is relatively small (in the analysis by Fotios, Robbins and Farrall, crimes were included if they occurred the week either side of each clock change, over a ten-year period, resulting in 40 weeks of crime counts included in their analysis). This limited time period can result in relatively low crime counts being used in the odds ratio calculation, producing large confidence intervals, particularly when analysed by crime category. An alternative approach that still uses the case / control method to isolate the effect of darkness is utilising a whole year approach - selecting a case hour that is in daylight for part of the year and in darkness for another part of the year. This is possible for locations that are sufficiently North or South of the equator to show a large seasonal variation in daylight hours. In Sheffield, United Kingdom, for example, the hour of 19:00–19:59 will be in darkness between January-March and October-December, but in twilight or daylight for the rest of the year. A control hour can also be selected that remains in daylight (or darkness) throughout the whole year. This method allows recorded crimes from across the whole year to be included in the odds ratio calculation, rather than just in a short window around each clock change, reducing the associated confidence intervals. This whole year approach has been applied in work assessing the effect of darkness on walking and cycling rates [30, 31], but not to assess the effect of darkness on crime rates.

Any influence of darkness on the risk of a crime occurring is unlikely to be equal for all types of crime. The committal of certain crimes may be more dependent on visibility levels than others, for example. Consider robbery - this is an interpersonal crime that requires the offender to get close to the victim. One factor that may deter a potential offender is the risk of being identified [32] - this risk reduces when it is dark and visibility is lower. The results of Fotios, Robbins and Farrall [27, 28] suggested darkness increased the number of robbery crimes but not other types of crimes. By contrast, burglary does not always rely on visibility levels - some residential dwellings are an excellent target in the daylight rather than after dark because of their unoccupied status [33]. Disaggregating crime counts by the type of crime reduces the sample included in the odds ratio calculations, and the relatively low counts in

some crime categories may account for why crimes other than robbery were not significantly affected by ambient light. Fotios, Robbins and Farrall [27] extrapolated their data from the three cities included in their analysis to calculate estimated odds ratios for the whole of the United States for different crime types. The extrapolated larger samples of crime counts suggested significant effects for other types of crime, such as destruction of property, theft and drunkenness. These conclusions were, however, based on extrapolated rather than real data.

The influence of darkness on crime risk may vary spatially. Certain streets or neighbourhoods may be more susceptible to after dark crime than others. This could be due to environmental factors - the physical form of the area, or a lack of adequate outdoor lighting, may reduce visibility after dark making some crimes more attractive to perpetrators. The risk of after-dark crime may also vary between areas due to the propensity of certain types of crime in those areas. If specific crimes are more likely to occur after dark than during daylight, those areas that show high levels of those crimes will also show a higher risk of crime after dark. Assessing the risk of crime occurring after dark for specific localised areas also has the added benefit that the area will act as a statistical control for itself, as other contributory factors that could act as confounds, such as aspects of the physical environment and the demographics of the area will remain constant.

In summary, there is much evidence that shows a relationship between light levels and perceptions of safety or reassurance. Evidence about the relationship between lighting and actual crime is not as clear. Whilst systematic reviews [14, 17] suggest 'improved lighting' (which can include an increase in the presence of lighting, such as illuminating a previously unlit route, or increases in brightness of lighting) is linked to reductions in crime there are statistical limitations to this evidence [15, 16]. The mechanism behind any effect of lighting on crime is also unclear, as much of the evidence that shows any effect does so for crimes during both daylight and after dark. This suggests improvements to visual conditions cannot be the only cause of an effect of lighting and other factors are at work, such as increases in community pride due to visible investment in an area. A first step in bringing clarity to the question of whether lighting influences crime levels is to assess the impact of darkness on crime, relative to daylight. If no effect is shown this rules out improvement of visual conditions as a causal mechanism for any effect of lighting on crime, and suggests there is no need to identify optimal lighting conditions for the reduction of crime after dark. If an effect is shown however, the size of this effect could be used as a measure to help optimise lighting characteristics such as illuminance and uniformity. Good lighting could be considered that which offsets any increase in criminal activity after dark.

Previous studies have shown that darkness can increase the risk of crime [26–28]. These works had limitations though such as a focus on less populous areas [26] and relatively limited sample sizes due to only using single weeks on either side of a clock change [27, 28]. There is also uncertainty about the timing of crimes included in the analysis of these previous works. Many crimes are aoristic, meaning an exact time of committal is not known, only a window of time within which it could have occurred. Knowing the timing of a crime relatively precisely is important when comparing crimes at the same period of the day under different ambient light conditions. The previous work by Doleac & Sanders, and Fotios, Robbins & Farrall, do not address this point and it is likely they included crimes that occurred outside the period of time they were targeting. Previous work has also not examined differences in the effects of light at a sub-district level, although Fotios, Robbins & Farrall do compare between cities.

In this study we assess the impact of darkness on crime levels using crime data for the South Yorkshire region of the United Kingdom, and we address some of the limitations from previous work highlighted above. We use the whole-year case / control method that controls for potentially confounding factors that may influence crime occurrence but are not related to

light levels, such as the time of day. This method increases the counts of crimes included in the analysis compared with the clock change method used by Doleac & Sanders [26] and Fotios, Robbins & Farrall [27, 28]. This will reduce uncertainty in odds ratios when the analysis is disaggregated by crime type. It will also reduce uncertainty when disaggregating by sub-district areas. We will also apply a screening process to ensure aoristic crimes are only included in the analysis if we can be relatively certain they occurred in one of our case or control periods.

We will test the following hypotheses:

1. The overall risk of crime in South Yorkshire occurring after dark is greater than during daylight, after time of day and seasonal factors have been accounted for

2. The risk of a crime occurring after dark relative to during daylight will vary depending on the type of crime

3. The risk of a crime occurring after dark relative to during daylight is not uniform across Middle Super Output Areas (defined below) within South Yorkshire

We will test these hypotheses by calculating odds ratios for all crime aggregated across the entire area of analysis (1), odds ratios for individual crime categories (2), and odds ratios for sub-areas within the entire area of analysis (3).

The next section outlines the planned method and analysis for testing the three hypotheses. As this work involves the analysis of secondary data, the template for preregistration of secondary data analysis provided by van den Akker et al [34] has also been completed and is included as S1 File.

## 2. Method

### 2.1 Definition of darkness and daylight

We define darkness as being when the sun's altitude is at or below -6˚. We choose this definition because this represents the transition between civil twilight (when the sun's altitude is between -6˚ and 0˚) and nautical twilight (when the sun's altitude is between -6˚ and -12˚ - see [35]). Based on data from solar monitoring sites in the UK, as reported in Raynham et al. [36], the average illuminance when the sun's altitude is at -6˚ is 2.33 lx.

We define daylight as being when the sun's altitude is at or above 0˚ [35]. This altitude represents the time of sunrise or sunset. Based on the solar illuminance data reported in Raynham et al [36], the average illuminance at this altitude is 509 lx.

The period when the sun's altitude is between -6˚ and 0˚ is defined as civil twilight and represents a transition between ambient daylight and ambient darkness.

### 2.2 Assessing the impact of darkness

To assess the impact of darkness on crime rates we can compare counts of crimes during periods of darkness against counts of crimes during periods of daylight. Darkness occurs at different times of the day to daylight however, and time of day acts as a significant confounding factor with this approach. We can therefore compare counts of crimes that occur within the same hour but across the whole year, choosing this hour so that for part of the year it is in darkness and part of the year it is in daylight. For example, in Sheffield (UK), the hour between 18:30 and 19:29 is entirely in darkness between 1st January and 6th March, and between 31st October and 31st December. Between the 31st March and 10th September the hour is entirely in daylight. For the remaining periods of the year at least part of the hour is in twilight. We can compare counts of crimes during this hour when it is in darkness with counts when it is in daylight.

Although the above approach removes time of day as a confounding factor, by using the same time of day for both periods of darkness and of daylight, it does not account for changes in weather conditions or other seasonal factors that may influence crime rates. For example, the period when the hour of 18:30–19:29 is in daylight also coincides with better weather conditions, compared with when the hour is in darkness. Other seasonal factors that could contribute to crime rates also vary between the periods of daylight and darkness. Holiday and vacation periods have previously been associated with changes in crime rates [37] and there is an extended school vacation during the daylight period. Therefore to account for seasonal factors that may influence crime rates, counts of crimes can also be recorded for a 'control' hour that remains in the same ambient light condition throughout the year. For example, for Sheffield, the hour of 14:00–14:59 remains in daylight between 1st January and 31st December. Changes between the crime counts occurring during the case hour when it is in daylight and darkness can be compared against changes in the crime counts occurring during the control hour during the same two periods of time. This comparison can be done by calculating an odds ratio, as shown in Eq 1, using four separate counts of crimes determined by whether they occurred in the case or control hour, and at what time in the year, as shown in Table 1. The odds ratio provides a measure of the effect of darkness on crime rates that accounts for both the time of day and other seasonal factors such as weather conditions and vacation periods. An odds ratio significantly greater than one indicates the risk of a crime occurring after dark is greater than during daylight. A confidence interval for the odds ratio can be calculated using Eq 2.

$$OddsRatio = \frac{CaseDark}{CaseDay} \div \frac{ControlDark}{ControlDay} \tag{1}$$

$$95\% \ Cl = exp\left(ln(OddsRatio) \ \pm \ 1.96\sqrt{\frac{1}{CaseDark} \ + \ \frac{1}{CaseDay} \ + \ \frac{1}{ControlDark} \ + \ \frac{1}{ControlDay}}\right) \tag{2}$$

Where:

*CaseDark* = Count of crimes in case hour when it is in darkness

*CaseDay* = Count of crimes in case hour when it is in daylight

*ControlDark* = Count of crimes in control hour when case hour is in darkness

*ControlDay* = Count of crimes in control hour when case hour is in daylight

For the current analysis three 60-minute periods will be selected as case hours and these will be paired with three control hours, as shown in Table 2.

## 2.3 Crime data

This analysis uses crimes recorded by South Yorkshire Police, whose jurisdiction covers the Local Authority areas of Barnsley, Sheffield, Rotherham and Doncaster in the United Kingdom.

**Table 1. Contingency table showing how the four counts used in the odds ratio calculation will be determined.**

| | Crimes that occurred on dates when the case hour is in darkness | Crimes that occurred on dates when the case hour is in daylight |
|---|---|---|
| **Crimes that occurred in the case hour** | *CaseDark* | *CaseDay* |
| **Crimes that occurred in the control hour** | *ControlDark* | *ControlDay* |

**Table 2. Pairs of case and control hours used for main analysis.**

| Case hour | Paired control hour |
|---|---|
| 17:30–18:29 | 13:00–13:59 |
| 18:30–19:29 | 14:00–14:59 |
| 19:30–20:29 | 15:00–15:59 |

The police record a crime through a variety of channels, most frequently through attending a report of an incident. A crime is submitted for recording either by the operator dealing with the incident or the officer attending the scene. The crime will then be recorded by the Force Crime Bureau (FCB), a sub division of the control room, and then allocated to an officer to investigate further. The crime is recorded on the force crime system, CONNECT (formerly CMS), according to the crime recording processes outlined in the 'Home Office counting rules for recorded crime' (HOCR). All police forces in the UK adhere to the HOCR and the National Crime Recording Standards (NCRS). This is to ensure crimes are recorded and counted in a standardised manner to allow for, amongst other reasons, a comparative overview of crime rates at a national and subnational level. In recording crime, there is oversight of the accuracy of this process conducted within South Yorkshire Police via external and internal audit provided in part by inspections conducted by His Majesty's Inspectorate of Constabulary (HMIC).

The data used in this analysis will be extracted from the CONNECT and CMS systems used by South Yorkshire Police to record crime incidents. Crimes recorded as taking place between 1st January 2010 and 31st December 2019 will be included in the analysis. This time period covers the operation of two crime recording systems in South Yorkshire Police, CMS (Jan 2010 –Nov 2017) and CONNECT (Dec 2017 –Dec 2019). In both cases data are extractable via the SQL based system of Oracle BI. The data contains information about the crime category recorded. This categorisation is based on a hierarchy of offences laid down in the HOCR and is detailed by HMIC in their crime tree diagram (see Fig 1). The study has opted to analyse crimes at the level 3 categorisation in the crime tree hierarchy (for the assessment of hypothesis 2), which strikes a balance of providing more detail to the nature of the crime versus the statistical power afforded/lost by aggregation/disaggregation.

Crimes that are classed as 'Other crimes against society' (also known as 'crimes against the state') will be excluded from the data. These crimes will be excluded as they can often be the result of police generated activity i.e. the result of recording crime that wouldn't have previously been reported because of proactive patrols/targeted interventions. An example of this is drug possession offences. These are often the result of a search or a targeted patrol and as such the more that activity occurs, the more the police record, meaning the indicator is a measure of police activity and not a base level of 'true' criminality.

The crime data also includes the Middle Super Output Area (MSOA) where each crime took place. MSOAs are a layer of geographical areas in the UK designed to support the reporting of small-area statistics. MSOAs have a minimum population of 5,000, with a mean population of 7,200. As the definition of an MSOA boundary is based around population level they can vary greatly in area - being relatively small in densely populated areas but large in sparsely populated areas. The analysis will provide overall crime odds ratios for each MSOA in South Yorkshire. Odds ratios by crime type for each MSOA will not be calculated because this level of disaggregation is likely to lead to very small crime counts included in the odds ratio calculation, and potentially also produces ethical issues in terms of anonymity, with small count data potentially allowing the identification of victims.

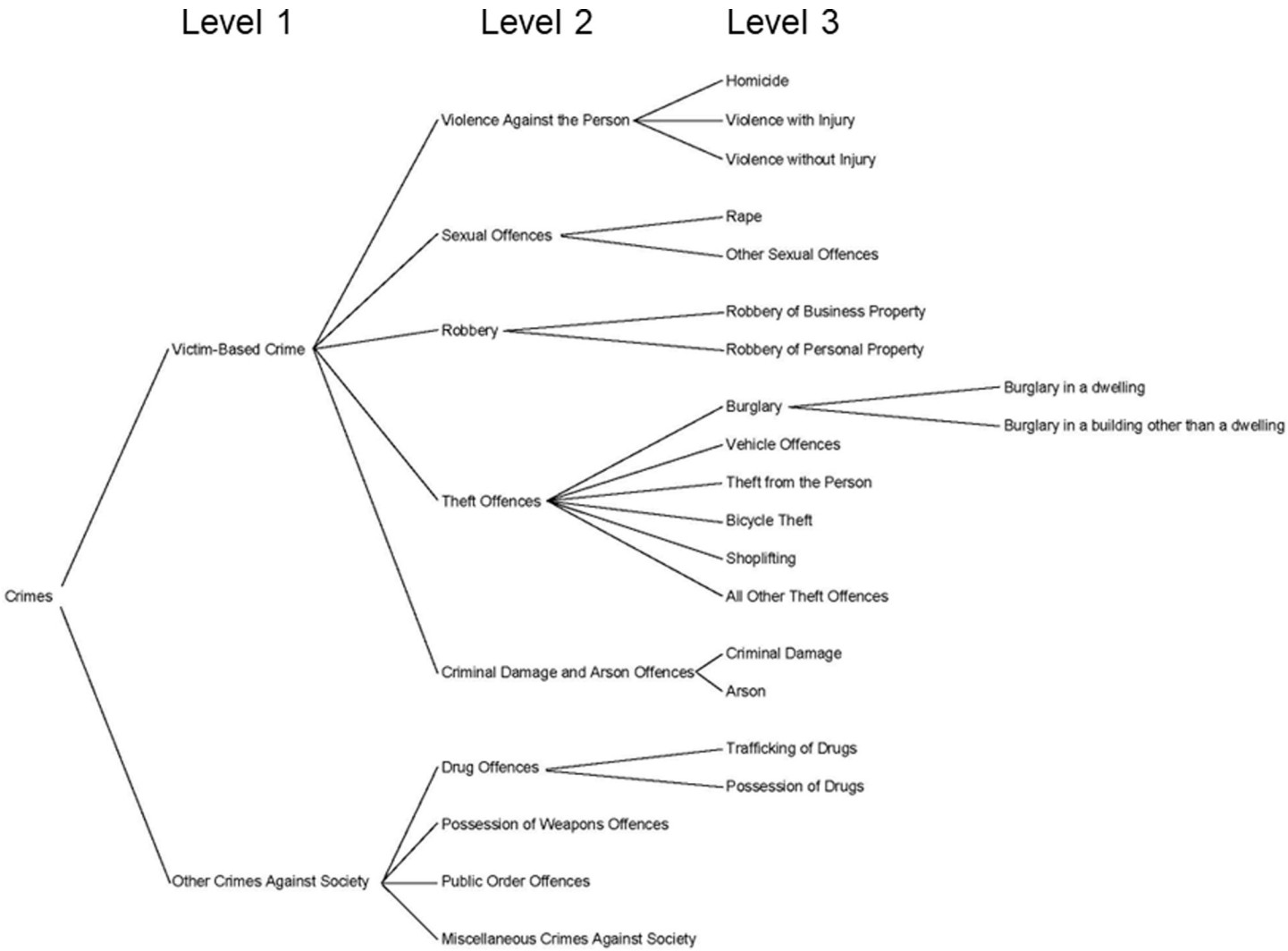

**Fig 1. The HMIC crime tree hierarchy with level 1, 2 and 3 categorisations.**

A fictional example of unfiltered and unprocessed data that will be used in this analysis is shown in Table 3.

**Table 3. Example of unfiltered, unprocessed fictional data.**

| Incident Number | HMIC Crime Tree Level 3 | MSOA | Committed From Date | Committed To Date | Start time | End time |
|---|---|---|---|---|---|---|
| 3 | ALL OTHER THEFT OFFENCES | Hackenthorpe | 22/09/2018 | 24/09/2018 | 17:12 | 03:41 |
| 4 | THEFT FROM THE PERSON | Handsworth South | 20/01/2018 | 22/02/2018 | 02:05 | 02:05 |
| 5 | BURGLARY - DWELLING | Lower Stannington | 23/03/2018 | 23/03/2018 | 18:12 | 21:53 |
| 6 | BICYCLE THEFT | Crabtree & Fir Vale | 31/10/2018 | 31/10/2018 | 16:56 | 19:00 |
| 7 | ROBBERY - BUSINESS | Southey Green West | 11/05/2018 | 11/05/2018 | 19:59 | 21:23 |
| 8 | ARSON | Sharrow | 14/02/2018 | 14/02/202018 | 18:22 | Not Recorded |

## 2.4 Data analysis

The analysis will calculate odds ratios to make inferences about the influence of darkness on the risk of crime. The data will first be filtered to only include those crimes that occur in pairs of case and control hours (see Table 2). The data includes information about the time and date the crime was committed. A 'Committed From Date' and 'Start time' are recorded. Our analysis script will combine these to create a 'committed from' time and date variable. A 'Committed To Date' and 'End time' are also recorded, and these will be combined to create a 'committed to' time and date variable. For crimes when the exact time of committal are known, these two time and date variables are identical. Some crime records do not include a value for 'End time', although the 'Committed To Date' is identical to the 'Committed From Date'. In these instances it will be assumed that the end time is identical to the start time. Many crimes are aoristic. For such crimes, the record of the offence provides a window of time the crime was potentially committed in. Crimes where the time of committal is known but have taken place over a period of time, rather than at a specific instance, may also have a window of time when the crime was committed. It is not possible to distinguish between these and aoristic crimes however, so we therefore treat both types of crime in the same way. Only crimes that could have been committed in a case or control hour should be included in the analysis. Therefore two inclusion criteria will be applied to aoristic crimes and crimes that are committed over a period of time, where an exact committal time is unknown, to determine their inclusion in the final dataset:

1. The midpoint between the 'committed from' and 'committed to' times should fall within one of the case or control hours (the midpoint within a crime time 'window' has been shown to be a better estimate of committal time than the start or end of that window [21])

2. The difference between the 'committed from' and 'committed to' times should be less than one hour

Crime records that meet these two inclusion criteria will be included in the final dataset for analysis. Crimes where the exact committal time is known (i.e. the 'committed from' and 'committed to' times are the same, or a 'committed from' time is given but not a 'committed to' time) will also be included in the final dataset if this time falls within one of the case or control hours.

Counts of those crimes that fall within a case or control hour, based on the above criteria, will be aggregated into four groups, based on when they occurred:

CaseDark: Crimes that occurred during one of the case hours when that case hour was in darkness.

CaseDay: Crimes that occurred during one of the case hours when that case hour was in daylight.

ControlDark: Crimes that occurred during one of the control hours when that hour's paired case hour was in darkness. For example, crimes that occurred during 13:00–13:59 would be included in this group if they occurred on a date when the paired case hour, 17:30–18:29, was in darkness.

ControlDay: Crimes that occurred during one of the control hours when that hour's paired case hour was in daylight.

Whether a case hour is in daylight or darkness will be defined by day of the year, as set out in Table 4. On days of the year that fall outside the ranges for darkness and daylight shown in Table 4 the case hour will partially be in twilight. Any crimes that occur in case or control hours on such dates will be excluded from the analysis.

**Table 4. Days of the year that define whether case hour is in daylight or darkness.** Day 1 represents 1st January, day 365 (or 366 if it is a leap year) represents 31st December.

| Case hour | Days of year case hour is in darkness | Days of year case hour is in daylight |
|---|---|---|
| 17:30–18:29 | 1–33, 304–365 (or 366 if leap year) | 87–278 |
| 18:30–19:29 | 1–65, 296–365 (or 366 if leap year) | 90–253 |
| 19:30–20:29 | 1–83, 270–365 (or 366 if leap year) | 120–227 |

Odds ratios will be calculated using counts for CaseDark, CaseDay, ControlDark and ControlDay. However, the odds ratio calculation requires all four of these counts to be non-zero. If one of the counts is zero, 0.5 will be added to all four counts used in the odds ratio, following the Haldane-Anscombe correction that is commonly used in such cases for odds ratios (e.g. see [38]). If more than one of the four counts is zero, the odds ratio will not be calculated.

To assess hypothesis 1, that the overall risk of crime occurring after dark is greater than during daylight, the counts of all crimes will be aggregated for each of the four periods and an overall odds ratio calculated from this, using Eq 1. The hypothesis will be confirmed if this odds ratio is significantly greater than 1.0, based on its associated p-value, calculated using Fisher's exact test, with a criterion for significance set at 0.05.

To assess hypothesis 2, that the risk of a crime occurring after dark relative to during daylight will vary depending on the type of crime, the counts of crimes for each crime category will be aggregated for the four periods and an odds ratio (using Eq 1) and associated 95% confidence interval (using Eq 2) will be calculated for each crime category. The hypothesis will be confirmed if any of the confidence intervals do not overlap.

To assess hypothesis 3, that the risk of a crime occurring after dark relative to during daylight is not uniform across MSOAs in South Yorkshire, the counts of all crimes will be aggregated for each of the four periods and a separate odds ratio and associated 95% confidence interval will be calculated for each MSOA. The hypothesis will be confirmed if any of the confidence intervals do not overlap.

The analytical script and two fictional datasets (one representing the CMS dataset and one representing the Connect dataset) to be used with this script are provided as S2–S4 Files. These fictional datasets have been generated randomly but reflect the structure, data types and variable headings that will be present in the real data. The script produces an output showing the sum of counts of crimes in different case and control periods, either for different crime categories or for different MSOAs, depending on how the 'location' variable is defined in the script. Table 5 shows the output for the fictional data by crime category, and Table 6 shows the output by MSOA.

These fictional results would not show support for hypothesis 1 as the odds ratio for all crime is not significantly above 1.0 (OR = 0.94, p > .05), suggesting the risk of crime after dark is not greater than during daylight. Hypotheses 2 and 3 would both be supported by the fictional results however. The confidence intervals for some crime categories do not overlap (for example, 'Robbery-Business' and 'Vehicle offences'), suggesting the risk of a crime occurring after dark compared with during daylight varies depending on the crime category. The confidence intervals for one MSOA, Woodhouse, does not overlap with those of some other MSOAs (Meersbrook, Shirecliffe and Southey), suggesting the risk of a crime occurring after dark compared with during daylight varies depending on the MSOA.

These results and conclusions are completely fictional at this stage, based on fictional, randomly generated date. An alternative conclusion to hypothesis 1 would be reached, that the risk of crime after dark is greater than during daylight, if the odds ratio for all crime was significantly greater than 1.0. Alternative conclusions to hypotheses 2 and 3 would also be reached if

**Table 5. Output from fictional data, by crime category.**

| Crime category | Count of crimes | | | | Odds Ratio | Lower confidence interval | Upper confidence interval |
| | CaseDark | CaseDay | ControlDark | ControlDay | | | |
| --- | --- | --- | --- | --- | --- | --- | --- |
| All other theft offences | 173 | 224 | 188 | 202 | 0.83 | 0.63 | 1.10 |
| Arson | 188 | 214 | 226 | 230 | 0.89 | 0.68 | 1.17 |
| Bicycle theft | 182 | 230 | 174 | 226 | 1.03 | 0.78 | 1.36 |
| Burglary | 201 | 219 | 203 | 217 | 0.98 | 0.75 | 1.29 |
| Criminal damage | 187 | 210 | 227 | 217 | 0.85 | 0.65 | 1.12 |
| Homicide | 204 | 212 | 184 | 238 | 1.24 | 0.95 | 1.63 |
| Other sexual offences | 207 | 204 | 209 | 210 | 1.02 | 0.78 | 1.34 |
| Possession of drugs | 202 | 244 | 185 | 213 | 0.95 | 0.73 | 1.25 |
| Rape | 180 | 220 | 197 | 198 | 0.82 | 0.62 | 1.09 |
| Robbery - business | 182 | 225 | 200 | 166 | 0.67 | 0.51 | 0.89 |
| Robbery - personal | 181 | 219 | 198 | 204 | 0.85 | 0.65 | 1.12 |
| Shoplifting | 172 | 234 | 213 | 189 | 0.65 | 0.49 | 0.86 |
| Stalking and harassment | 175 | 195 | 179 | 181 | 0.91 | 0.68 | 1.21 |
| Theft from the person | 187 | 210 | 172 | 215 | 1.11 | 0.84 | 1.47 |
| Trafficking of drugs | 197 | 221 | 183 | 205 | 1.00 | 0.76 | 1.32 |
| Vehicle offences | 206 | 217 | 165 | 234 | 1.35 | 1.02 | 1.77 |
| Violence with injury | 168 | 202 | 190 | 236 | 1.03 | 0.78 | 1.37 |
| Violence without injury | 198 | 195 | 202 | 207 | 1.04 | 0.79 | 1.37 |
| ALL CRIME | 3390 | 3895 | 3495 | 3788 | 0.94 | 0.88 | 1.01 |

all of the odds ratio confidence intervals overlapped for the different crime categories, and for the different MSOAs. Such results would suggest crime categories did not differ in their risk of being committed after dark compared with during daylight (lack of support for hypothesis 2), and MSOAs did not differ in their risk of crime being committed after dark compared with during daylight (lack of support for hypothesis 3).

## 2.5 Sensitivity analysis

We have chosen three pairs of case and control hours to use in this analysis. It is possible the results will be sensitive to these choices, and selecting different pairs of case and control hours could yield different conclusions in relation to the three tested hypotheses. To assess this sensitivity to choice of case and control hours we will also carry out a sensitivity analysis by using two alternative case hours from the morning, 05:00–05:59 and 06:00–06:59, paired with the control hours of 11:00–11:59 and 12:00–12:59 respectively. The days of the year that the two alternative case hours will be in daylight or darkness are shown in Table 7. This sensitivity analysis will help show whether the odds ratios reported in the main analysis are sensitive to the choice of case and control hours. This could also help show whether the time of day influences how darkness affects crime. Different types of crime tend to occur at different times of

**Table 6. Output from fictional data, by MSOA.**

| MSOA name | Count of crimes | | | | Odds Ratio | Lower confidence interval | Upper confidence interval |
|---|---|---|---|---|---|---|---|
| | CaseDark | CaseDay | ControlDark | ControlDay | | | |
| Broomhill | 201 | 221 | 196 | 191 | 0.89 | 0.67 | 1.17 |
| Chapeltown | 196 | 201 | 193 | 214 | 1.08 | 0.82 | 1.43 |
| City Centre | 177 | 211 | 190 | 210 | 0.93 | 0.70 | 1.23 |
| Dore | 208 | 220 | 217 | 227 | 0.99 | 0.76 | 1.29 |
| Ecclesfield | 182 | 230 | 193 | 222 | 0.91 | 0.69 | 1.20 |
| Heeley | 183 | 243 | 205 | 221 | 0.81 | 0.62 | 1.06 |
| High Green | 184 | 209 | 197 | 203 | 0.91 | 0.69 | 1.20 |
| Hillsborough | 179 | 227 | 195 | 223 | 0.90 | 0.69 | 1.19 |
| Lowedges | 203 | 238 | 206 | 202 | 0.84 | 0.64 | 1.10 |
| Meersbrook | 192 | 214 | 180 | 223 | 1.11 | 0.84 | 1.47 |
| Mosborough | 191 | 219 | 198 | 212 | 0.93 | 0.71 | 1.23 |
| Parson Cross | 175 | 210 | 170 | 215 | 1.05 | 0.79 | 1.40 |
| Shirecliffe | 204 | 205 | 178 | 210 | 1.17 | 0.89 | 1.55 |
| Shiregreen | 178 | 189 | 188 | 194 | 0.97 | 0.73 | 1.29 |
| Southey | 190 | 201 | 186 | 227 | 1.15 | 0.87 | 1.52 |
| Totley | 188 | 244 | 192 | 219 | 0.88 | 0.67 | 1.15 |
| Wisewood | 195 | 197 | 202 | 194 | 0.95 | 0.72 | 1.26 |
| Woodhouse | 0.5 | 4.5 | 4.5 | 0.5 | 0.01 | 0.00 | 0.77 |
| Woodseats | 164 | 212 | 205 | 181 | 0.68 | 0.51 | 0.91 |
| ALL CRIME | 3390 | 3895 | 3495 | 3788 | 0.94 | 0.88 | 1.01 |

day [24] due to variation in structural opportunities to commit crime at different hours of the day. As well as being a validation of the main analysis, the sensitivity analysis will help show whether crimes committed at different hours are affected by darkness in different ways.

## 2.6 Statistical power

The epi.ssc function from the *R* package *epiR* was used to estimate the total count of crimes in the case and control periods required to detect odds ratios of sizes ranging between 1.2 and 2.5, with a minimum power of 80%, a confidence level of 95%, and a one-sided test. These are shown in Table 8. In calculating these required counts the following assumptions were made, based on the data reported in Fotios et al [27]:

**Table 7. Days of the year that define whether alternative case hours for sensitivity analysis are in daylight or darkness.**

| Case hour | Days of year case hour is in darkness | Days of year case hour is in daylight |
|---|---|---|
| 05:00–05:59 | 1–67, 90–91, 257-365/366 | 141–196 |
| 06:00–06:59 | 1–39, 291–297, 325-365/366 | 84, 110–234 |

**Table 8. Required total counts in case and control hours to detect different odds ratios with 80% power.**

| Odds ratio | Count required in case hour | Count required in control hour |
|---|---|---|
| 1.2 | 1070 | 2461 |
| 1.5 | 219 | 504 |
| 1.8 | 105 | 242 |
| 2.1 | 67 | 155 |
| 2.4 | 49 | 113 |

1. The proportion of crimes in the control hour when it was in darkness was 50% of all crimes recorded in the control hour

2. The total count of crimes in the control hour was 2.3 times that of the total count of crimes in the case hour

We do not yet know how much data will be available to use in our analysis and what the frequency of counts in case and control hours will be, as access to the dataset has been deliberately withheld (see section 2.7). We can however make an educated estimate. Between 2009/10 and 2018/19 there were 990,446 crimes recorded in South Yorkshire [39], excluding those categorised as 'Other crimes against society'. Therefore 41,269 crimes were committed, on average in each hourly period across the entire 10-year period. A large proportion of these crimes will have a potential time range of when they could have been committed of more than an hour and would therefore not be included in our analysis. Data from Ratcliffe [40] suggests 51% of crimes have a time window of less than 4 hours. We therefore assume that 20% of crimes will have a time window of one hour or less, and will be included in our analysis, although this assumption is not based on any specific rationale. This would leave a count of 8,254 crimes in each hour, on average, over 10 years. We will include counts from six different hourly periods in our main analysis (three case hours and three control hours), giving a total count of 49,524. Based on the ratio of 2.3 crimes in a control hour for every crime in the case hour as found by Fotios et al [27], this suggests there could be 15,008 crimes in the case hours and 34,516 crimes in the control hours. Such counts allow an odds ratio of 1.05 or greater to be detected.

## 2.7 Prior knowledge of data

Prior knowledge of the data to be used in this analysis has deliberately been limited. This is to avoid the opportunity to p-hack the data and obtain statistically significant but spurious results [41], and to limit the formulation of hypotheses that are already known to be in line with the data (Hypothesising After the Results are Known - HARKing - see [42]). Access to the data is provided by three of the authors (JS, SF and RC) who are employees of South Yorkshire Police and have access to the crime records database from which data will be extracted. However, the hypotheses and analytical approach have been developed by two of the authors (JU and SAF) who have no direct access to or prior experience of the data to be used. An analytical script will be written in *R* by JU and SAF, and this will be passed to JS, SF and RC to apply to the crime data.

## Supporting information

**S1 File. Preregistration template for analysis of secondary data.**
(DOCX)

**S2 File. R code for analysis of crime data.**
(TXT)

**S3 File. Fictional crime data (from CMS system).**
(CSV)

**S4 File. Fictional crime data (from Connect system).**
(CSV)

## Acknowledgments

The authors wish to acknowledge the support and contribution made to this work by Ray Froggatt, South Yorkshire Police.

## Author Contributions

**Conceptualization:** Jim Uttley.

**Data curation:** Rosie Canwell, Jamie Smith, Sarah Falconer.

**Formal analysis:** Jim Uttley, Rosie Canwell, Sarah Falconer.

**Methodology:** Jim Uttley.

**Project administration:** Jim Uttley, Jamie Smith.

**Resources:** Jamie Smith.

**Supervision:** Jamie Smith, Steve A. Fotios.

**Validation:** Jim Uttley.

**Writing – original draft:** Jim Uttley, Steve A. Fotios.

**Writing – review & editing:** Jim Uttley, Rosie Canwell, Jamie Smith, Sarah Falconer, Yichong Mao, Steve A. Fotios.

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
