## [Decision Letter · Decision Letter 0]

29 May 2023

PONE-D-23-08739Does darkness increase the risk of certain types of crime? A Registered Report ProtocolPLOS ONE

Dear Dr. Jim,

Thank you for submitting your manuscript to PLOS ONE. After careful consideration, we feel that it has merit but does not fully meet PLOS ONE’s publication criteria as it currently stands. Therefore, we invite you to submit a revised version of the manuscript that addresses the points raised during the review process.

Please revise the manuscript following the points raised by the reviewer.

We look forward to receiving your revised manuscript.

Kind regards,

Manuel Spitschan

Academic Editor

PLOS ONE

Journal Requirements:

2. In your cover letter, please confirm that the research you have described in your manuscript, including participant recruitment, data collection, modification, or processing, has not started and will not start until after your paper has been accepted to the journal (assuming data need to be collected or participants recruited specifically for your study). In order to proceed with your submission, you must provide confirmation.

Reviewers' comments:

Reviewer's Responses to Questions

**Comments to the Author**

1. Does the manuscript provide a valid rationale for the proposed study, with clearly identified and justified research questions?

Reviewer #1: Yes

Reviewer #2: Yes

2. Is the protocol technically sound and planned in a manner that will lead to a meaningful outcome and allow testing the stated hypotheses?

Reviewer #1: Partly

Reviewer #2: Yes

3. Is the methodology feasible and described in sufficient detail to allow the work to be replicable?

Reviewer #1: Yes

Reviewer #2: Yes

4. Have the authors described where all data underlying the findings will be made available when the study is complete?

Reviewer #1: Yes

Reviewer #2: No

5. Is the manuscript presented in an intelligible fashion and written in standard English?

Reviewer #1: Yes

Reviewer #2: Yes

6. Review Comments to the Author

You may also provide optional suggestions and comments to authors that they might find helpful in planning their study.

Reviewer #1: I think this is a really interesting project, and I particularly like the granularity of the analysis by looking at different types of crimes rather than an aggregated assessment of the effect of lighting on crime.

However, there is one primary methodological concern that would make it difficult to interpret the results and understand the theory of change.

I do not think the methodology is sufficiently rigorous because looking at daytime, evening, and nighttime comparisons misses the important attributes of crime, including the time of offence literature. For example, we know that crime concentrates in the afternoon hours because juveniles come out of school and interact with other juveniles. But the ones that commit crimes at night are different. We also know that there are different structural opportunities to commit crime at different hours of the day, depending on, for example, the availability of victims. We have also learned that there are hotspots of crime that are affected by ecological factors and are conditional on the time of day (i.e., shopping centres or cars to break into). In short, the methodology will miss all that.

There is not much that can be done by way of statistical controls on the same units. Instead, attempt to find areas that are as close as possible to one another where some have lighting and some do not, and if you can then control for confounding variables through randomization, even better. If randomization cannot be achieved, then (a) find hotspots of crime; (2) separate them by light/no light or different levels of light or perhaps different types of lights; and then try to do propensity score matching or some other technique to contrast the two groups.

I hope this helps.

Reviewer #2: # Summary

Solid and clearly explained study.

The background justifies the study.

The planned analysis should deal with the potential confounds.

# Important

I would like to see the outputs of the analysis of the fictional data, ideally in the main text. Ideally, with multiple sets of fictional data that would represent a range of support for/against the various hypotheses.

S1-5.3 - how does this compare with the total amount of data available? What range of odds ratios do you have power to detect?

Might want to bump/copy this section to main - it's important

"data collected during the study will be made fully available without restriction upon study completion" - I think this requires clarification. If I've understood correctly, the raw data will not be made available. What will be?

"Level 3" needs to be defined in text. Perhaps bump S2 to main?

S1, eq2 is broken

---

# Minor

"reduced significantly less" - confusing wording

I'm surprised there isn't any info considering the issue from the perspective of offenders in terms of how light effects motivation/ability to commit crime. Is there a literature on this?

> If the risk of crime occurring is shown to be higher after dark than in daylight in an area, effective lighting for crime reduction would reduce this relative increase in risk after dark.

I think this statement needs to acknowledge remaining potential confounds as caveats

It might be worth considering adding a contingency table pre-eq1, and using the standard nomclomenture in eq1 (OR = AD/BC).

Example OR values and associated conclusions, for demonstration purposes?

"the case hours being" - just ref table 1 rather than repeat?

Aoristic defined twice

Seems to assume that all crimes occur in less than 1 minute - isn't some of the difference between start and end time would be the length of time that the crime was taking place, rather than just uncertainty?

L451 might want to note that 0.05 implemented by 1.96 in eq2

7. PLOS authors have the option to publish the peer review history of their article (what does this mean?). If published, this will include your full peer review and any attached files.

Reviewer #1: No

Reviewer #2: **Yes: **Daniel Garside

---

## [Author Response · Author response to Decision Letter 0]

11 Aug 2023

Please see 'Response to reviewers' document table that is included with the revised manuscript.

---

## [Decision Letter · Decision Letter 1]

10 Sep 2023

Does darkness increase the risk of certain types of crime? A Registered Report Protocol

PONE-D-23-08739R1

Dear Dr. Uttley,

We’re pleased to inform you that your manuscript has been judged scientifically suitable for publication and will be formally accepted for publication once it meets all outstanding technical requirements.

Kind regards,

Manuel Spitschan

Academic Editor

PLOS ONE

Reviewers' comments:

Reviewer's Responses to Questions

**Comments to the Author**

1. Does the manuscript provide a valid rationale for the proposed study, with clearly identified and justified research questions?

Reviewer #2: Yes

2. Is the protocol technically sound and planned in a manner that will lead to a meaningful outcome and allow testing the stated hypotheses?

Reviewer #2: Yes

3. Is the methodology feasible and described in sufficient detail to allow the work to be replicable?

Reviewer #2: Yes

4. Have the authors described where all data underlying the findings will be made available when the study is complete?

Reviewer #2: Yes

5. Is the manuscript presented in an intelligible fashion and written in standard English?

Reviewer #2: Yes

6. Review Comments to the Author

You may also provide optional suggestions and comments to authors that they might find helpful in planning their study.

Reviewer #2: Thank you for responding to my comments and suggestions.

I am now satisfied that this study should be able to complete the stated goals.

7. PLOS authors have the option to publish the peer review history of their article (what does this mean?). If published, this will include your full peer review and any attached files.

Reviewer #2: **Yes: **Daniel Garside

---

## [Editor Report · Acceptance letter]

9 Jan 2024

PONE-D-23-08739R1 

PLOS ONE

Dear Dr. Uttley, 

I'm pleased to inform you that your manuscript has been deemed suitable for publication in PLOS ONE. Congratulations! Your manuscript is now being handed over to our production team.

Kind regards, 

on behalf of

Dr. Manuel Spitschan 

Academic Editor

PLOS ONE